# Use of Pulse Oximetry during Resuscitation of 230 Newborns—A Video Analysis

**DOI:** 10.3390/children10071124

**Published:** 2023-06-28

**Authors:** Vilde Kolstad, Hanne Pike, Joar Eilevstjønn, Frederikke Buskov, Hege Ersdal, Siren Rettedal

**Affiliations:** 1Department of Research, Stavanger University Hospital, 4019 Stavanger, Norway; vilde.hagensen.kolstad@sus.no (V.K.); frederikke.heinz.buskov@sus.no (F.B.); 2Department of Paediatrics, Stavanger University Hospital, 4019 Stavanger, Norway; hanne.markhus.pike@sus.no; 3Laerdal Medical, 4007 Stavanger, Norway; joar.eilevstjonn@laerdal.com; 4Faculty of Health Sciences, University of Stavanger, 4021 Stavanger, Norway; hege.ersdal@safer.net

**Keywords:** pulse oximetry, newborn saturation, heart rate assessment, heart rate monitoring, neonatal resuscitation, newborn resuscitation, resuscitation guidelines, positive pressure ventilation, NeoBeat

## Abstract

Background: European guidelines recommend the use of pulse oximetry (PO) during newborn resuscitation, especially when there is a need for positive pressure ventilation or supplemental oxygen. The objective was to evaluate (i) to what extent PO was used, (ii) the time and resources spent on the application of PO, and (iii) the proportion of time with a useful PO signal during newborn resuscitation. Methods: A prospective observational study was conducted at Stavanger University Hospital, Norway, between 6 June 2019 and 16 November 2021. Newborn resuscitations were video recorded, and the use of PO during the first ten minutes of resuscitation was recorded and analysed. Results: Of 7466 enrolled newborns, 289 (3.9%) received ventilation at birth. The resuscitation was captured on video in 230 cases, and these newborns were included in the analysis. PO was applied in 222 of 230 (97%) newborns, median (quartiles) 60 (24, 58) seconds after placement on the resuscitation table. The proportion of time used on application and adjustments of PO during ongoing ventilation and during the first ten minutes on the resuscitation table was 30% and 17%, respectively. Median two healthcare providers were involved in the PO application. Video of the PO monitor signal was available in 118 (53%) of the 222 newborns. The proportion of time with a useful PO signal during ventilation and during the first ten minutes on the resuscitation table was 5% and 35%, respectively. Conclusion: In total, 97% of resuscitated newborns had PO applied, in line with resuscitation guidelines. However, the application of PO was time-consuming, and a PO signal was only obtained 5% of the time during positive pressure ventilation.

## 1. Introduction

Approximately 5% of newborns require resuscitation with positive pressure ventilations (PPV) at birth [1,2,3,4]. According to resuscitation guidelines, heart rate (HR) should be assessed after birth to evaluate the transition and identify newborns in need of resuscitation [5]. PPV should be initiated within 60 s if the newborn has not established effective breathing and HR is <100 beats per minute (bpm) and not increasing in response to initial drying and stimulation. Auscultation by stethoscope is an inexpensive and simple method that provides a rapid and intermittent assessment of HR, but HR is sometimes underestimated [2,6]. Continuous monitoring with pulse oximetry (PO) ± electrocardiogram (ECG) has the advantage of providing a more dynamic indication of HR changes and information on the responses to resuscitative interventions [5]. Several studies have shown that ECG presents HR more rapidly than PO, and that PO may underestimate HR in the initial minutes [7,8,9,10,11,12]. The Consensus on Science with Treatment Recommendations (CoSTR) from the International Liaison Committee on Resuscitation (ILCOR) therefore recommend the use of ECG for newborn HR assessment during delivery room resuscitation whenever possible [3,5]. Where ECG is not available, HR assessment with PO is a reasonable alternative, but the limitations should be kept in mind.

ECG does not replace the need for PO for evaluation of oxygenation and subsequent titration of oxygen to avoid hyper- and hypoxia [2,11]. International guidelines recommend the use of PO during PPV or when providing supplemental oxygen [2,3], and PO is common practice during newborn resuscitation in many high-resource settings [13].

The objectives were to evaluate (i) to what extent PO was used during newborn resuscitations; (ii) the number of healthcare providers (HCPs) involved, the number of single-use sensors required for the application of PO, time spent on application or adjustments of PO; and (iii) the proportion of time with a PO signal during provision of positive pressure ventilation at birth.

## 2. Materials and Methods

### 2.1. Setting

A prospective observational study was conducted at Stavanger University Hospital, Norway from 6 June 2019 to 16 November 2021. The hospital is well suited for population-based studies, being the only hospital in the region with obstetric and neonatal services and 4200 annual deliveries. Vaginal births take place in the labour ward and the midwife-run low-risk ward, and the caesarean sections in the operating theatre. PPV is provided to 3.6% of newborns, mainly by flow-driven t-piece resuscitator (NeoPuff, Fischer and Paykel, Auckland, New Zealand), alternatively a self-inflating silicone resuscitator (Laerdal, Stavanger, Norway) [4]. The paediatric resident and midwife initiate anticipated resuscitations, with the addition of a neonatologist, neonatal nurse, and an anaesthetic team for advanced resuscitations. The national resuscitation guidelines derive from the European Resuscitation Council Guidelines [2].

### 2.2. Data Collection and Equipment

Video recordings of resuscitations were obtained passively using motion-triggered cameras placed above the resuscitation tables, capturing the newborn and the hands of the HCPs. Dry-electrode ECG (NeoBeat, Laerdal Medical, Stavanger, Norway) was placed on the chest or upper abdomen of the newborn by the midwife assistants as soon as possible after birth. NeoBeat displays and stores HR within seconds of birth and provide continuous HR measurements. If a newborn required PPV, the cord was clamped and cut, and the newborn was carried to the resuscitation table. HCPs were instructed to apply 3-lead gel-electrode ECG (CareFusion, San Diego, CA, USA) on the newborns’ chest and PO (Massimo LNCS Neo wraparound sensor, Massimo, Irvine, CA, USA) on the newborns’ right hand or wrist. The order of ECG and PO placement was left to the discretion of the HCPs. However, in anticipated events, one HCP was assigned the task of applying the PO sensor. The monitor used was Carescape Patient monitor B450 or B650 (GE Healthcare, Boston, MA, USA). Video recordings were used to evaluate the time from birth to the start of ventilation, duration of PPV, application of PO and ECG, and to detect HR and PO signals displayed on the monitor. Extraction of patient characteristics were automated from electronic medical records.

### 2.3. Inclusion Criteria

Parents were informed and consented to participation during routine follow up in pregnancy. Newborns born at gestational age (GA) ≥ 28 weeks who required resuscitation with PPV at birth with the resuscitation captured on video, were enrolled.

### 2.4. Calculations and Definitions

All videos were manually reviewed by independent researchers (V.K. and S.R. or H.P.). Timelines were started when the newborn was placed on the resuscitation table, and for a duration of ten minutes. Videos were annotated using the ELAN 5.8 tool (The Language Archive, Nijmegen, The Netherlands). The start of PPV was defined as when the first inflation was provided. The duration of PPV was defined as the time difference between the first and last inflation. The number of HCPs involved in applying the PO equipment and the number of single-use PO sensors used were recorded. The time spent applying the PO was defined from when one started drying the skin or picking up the sensor, whichever came first, until the PO was attached, regardless of whether a useful signal was obtained. The PO adjustment time included repositioning or reapplying the sensor, reattaching the PO cord to the monitor and/or changing the sensor. The PO signal was annotated based on reviewing a recording of the monitor screen. The time to reliable PO signal was defined as the time when a continuous pulse wave, HR, and/or saturation values were displayed for at least 3 s [7,12,14].

### 2.5. Statistical Analysis

Data and annotations were extracted and analysed using Matlab R2022b (MathWorks Inc., Natick, MA, USA). Continuous variables are expressed as median (quartiles) or count (%).

### 2.6. Ethical Considerations

This study was a part of the Safer Births Stavanger research project on newborn resuscitation, with ethical approval from the Norwegian regional ethical committee 27 April 2018 (ref.2018/338).

## 3. Results

In total, 10,539 newborns were born during the study period, of which 10,503 with gestational age ≥28 weeks. Of these, 7466 (71%) parents agreed to participation, of which 289 (3.9%) received PPV at birth. The resuscitation was captured on video in 230/289 cases, and these newborns were included in the analysis. A flow diagram of over participants and patient characteristics is shown in Figure 1 and Table 1, respectively.

### 3.1. Use of Auscultation, ECG and PO Assessment during Resuscitation

Among 230 resuscitated newborns, 222 (97%) had PO applied during the first 10 min on the resuscitation table. PO was applied median (quartiles) 60 (24, 58) seconds after placement on the resuscitation table.

Auscultation with the stethoscope was performed in 211 (92%) median 39 (10, 110) seconds after placement on the table.

ECG was applied in 220 (96%) newborns. Standard 3-lead gel-electrode ECG was placed in 192 newborns at a median of 60 (41, 109) seconds. Dry-electrode ECG was in most instances placed in the delivery room, and the time of placement of NeoBeat was therefore median 0 (0, 3) seconds from placement on the resuscitation table (*n* = 204).

### 3.2. Number of HCPs Involved and Equipment Used for Application and Adjustment of PO

For 40% of newborns, one HCP was involved in the placement of PO. For 32% and 22% of newborns, two and three HCPs were involved, respectively. For the remaining 6%, four to six HCPs were at some time involved in the placement of PO. In 82% of the 222 resuscitations, one single-use sensor was used, whereas two sensors were used in the remaining 18% of cases.

### 3.3. Time Spent on the Application and Adjustment of PO

The median time to start the application of PO was 60 (23, 118) seconds after the newborn was placed on the resuscitation table. Time spent on applying and adjusting PO in the first 10 min of resuscitation was median 99 (44, 223) seconds. The time spent placing the PO is illustrated in blue lines in Figure 2. The proportion of time used on application and adjustments of PO during PPV and during the first ten minutes was 30% and 17%, respectively. In 58 (26%) newborns assessed with PO, the application was not completed by the time PPV ended.

### 3.4. Feedback on PO Signal

Video of the PO monitor signal was available in 118 (53%) of the 222 newborns. In 110 of 118 newborns a PO signal was obtained within the first ten minutes on the resuscitation table. The time when a PO signal was first displayed and the proportion of time with a PO signal in relation to PPV is shown in Figure 3a and b, respectively. The median time to obtain PO signal was 238 (129, 324) seconds. PPV started a median of 21 (11, 51) seconds after placement on the resuscitation table, and lasted for a duration of 126 (65, 232) seconds (data available for 222 newborns). In 85 of 118 of newborns, PO was applied during PPV. Among these, a median time with a PO signal displayed during the provision of PPV was 9 (0, 84) seconds. The proportion of time with a PO signal during PPV and during the first ten minutes after placement on the resuscitation table was 5% (0%, 31%) and 35% (9%, 49%), respectively.

## 4. Discussion

In this population-based video study of 230 newborns receiving PPV at birth, we found that PO was applied in 97% of resuscitations, in line with guideline recommendations [2,5]. However, HCPs spent 30% of the active resuscitation with PPV applying and adjusting the PO sensor, and median two HCPs were involved. Throughout the duration of PPV, a PO signal was only displayed 5% of the time.

An accurate HR assessment is considered important to evaluate the newborn condition, to guide management, and to evaluate the effect of resuscitative interventions. PO has several limitations with regard to monitoring HR in the delivery room. Our group has recently shown that PO displayed HR values later and for a shorter proportion of newborn resuscitation, when compared to the dry-electrode ECG device NeoBeat or standard three-lead ECG [15]. PO may furthermore underestimate HR signal for the first five to six minutes during newborn transition or resuscitation, especially in the more compromised newborns [12,15,16,17].

In the current study, the median time to obtain a PO signal was 237 s after placement on the resuscitation table. Similar findings have been reported previously, with a success rate in obtaining saturation measurements varying between 20–100% at one minute after birth [18]. Several studies have found that reliable PO signals are rarely available in the first two minutes after birth [6,7,8,19,20]. A previous study by our team demonstrated that PO signal may be delayed in newborns with low APGAR scores, who represent the group in most need of HR feedback [7]. Others have evaluated the order of sensor application to the newborn with respect to the acquisition of a reliable PO signal [21,22,23,24], and video recordings in the present study showed that the practise did vary. The type of PO sensor used can also affect the time to a reliable signal during newborn resuscitations [25].

The results challenge the role of PO in the assessment of HR during newborn resuscitations. PO does, however, have an important role in the titration of oxygen provision during resuscitation in order to avoid hypoxia and hyperoxia with oxidative stress [25]. Although none of our videos showed a sign that PO disrupted or delayed PPV, the effort of placing PO rarely provided useful signal on HR or tissue oxygenation during ventilation. Most resuscitations in our setting were of short duration with moderately asphyxiated newborns. PO may have a more prominent role in prolonged and advanced resuscitations or stabilization of preterms. PO is also used to confirm if the newborn has the capacity to maintain stable oxygenation within the recommended range [26], and may help identify newborns in need of admission to the neonatal intensive care unit.

Considerable resources went into the placement of PO. HCPs spent 30% of the time during active resuscitation applying PO, without obtaining PO signal in the majority of cases. Where resources are limited, prioritizing the placement of PO during ventilation may be impractical.

This study was limited by the loss of PO signal data in 104 newborns due to technical errors, including frame freezing of the monitor video and loss of resuscitation videos due to a server failure without a backup. In this study, we analysed the availability of PO signal during resuscitation, but have not evaluated the HR signal accuracy compared to the gold standard ECG. The population was limited to newborns with gestational age ≥28 weeks, since extremely premature newborns were resuscitated and stabilized in incubators not equipped with video cameras. The strengths of the study are the population-based design and a high number of included newborns. The setting is representative of high-resource contexts.

## 5. Conclusions

During delivery room resuscitation, 97% of newborns had PO applied in line with European Resuscitation Council Guideline recommendations. However, the application of PO was time-consuming, and PO signal were only displayed for 5% of the time of active resuscitation with PPV.

## Figures and Tables

**Figure 1 children-10-01124-f001:**
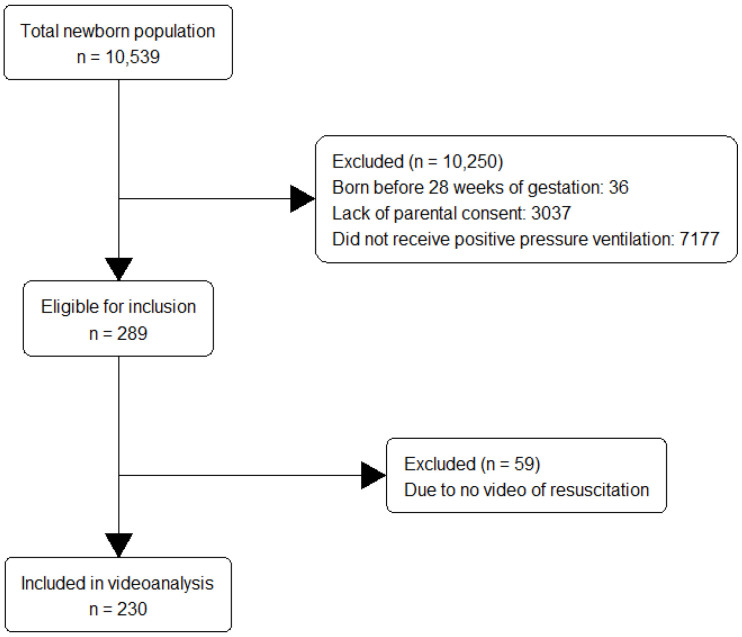
A flow diagram of over participants.

**Figure 2 children-10-01124-f002:**
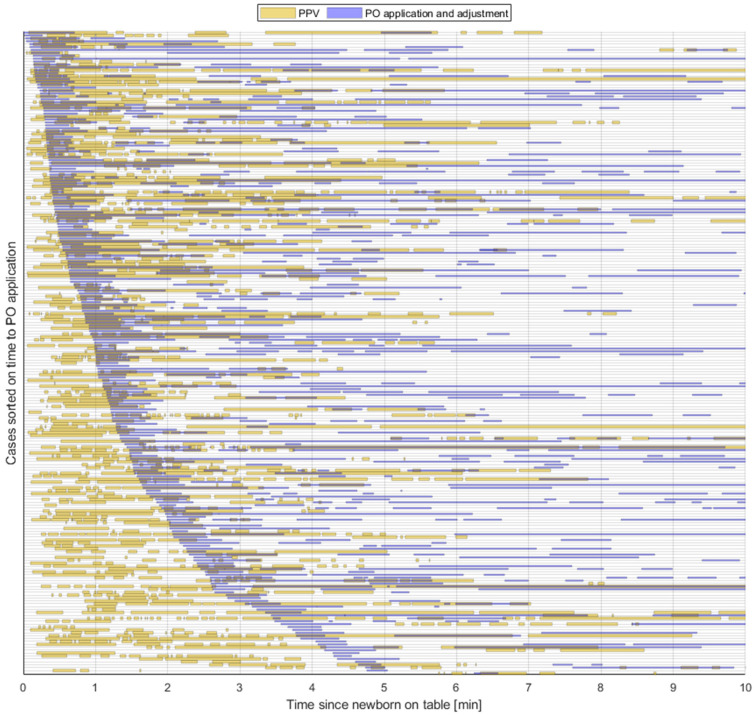
The figure shows 222 newborns that received PPV at birth and had PO applied. Each horizontal line represents a newborn from the time of placement on the resuscitation table (time = 0) and the first consecutive ten minutes. Provision of PPV is illustrated in orange lines. Application and/or adjustment of PO is illustrated in blue lines. Newborns are sorted by time from placement on the resuscitation table to start application of PO. PPV = positive pressure ventilation, PO = pulse oximetry.

**Figure 3 children-10-01124-f003:**
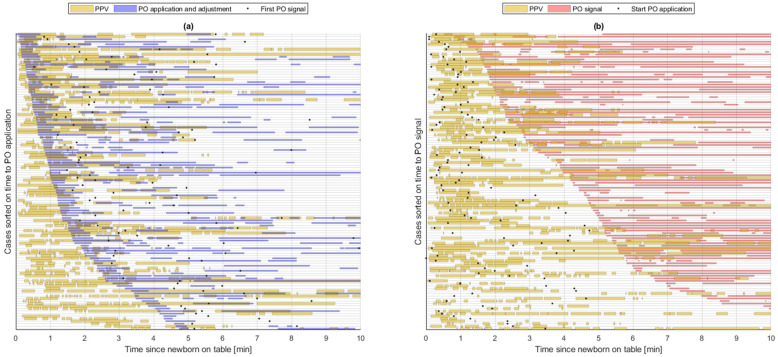
The figure shows the 118 newborns where video of the PO monitor was available. Each horizontal line represents a newborn from the time of placement on the resuscitation table and throughout the first ten minutes. (**a**) shows the application and adjustment of PO illustrated in blue lines in relation to PPV in orange lines, and time when the first PO signal was displayed illustrated as a black dot. (**b**) shows the proportion of time with a PO signal illustrated in pink lines in relation to PPV illustrated in orange lines. The black dot represents the start of PO application. PPV = positive pressure ventilation, PO = pulse oximetry.

**Table 1 children-10-01124-t001:** Characteristics of 230 newborns included in the analysis. Characteristics are reported as median (quartiles) or count (%).

Newborn Characteristics n = 230	
Variable	
Gestational age (weeks)	40 (38, 40)
Very preterm (28 to <32 weeks)	5 (2.2%)
Moderate preterm (32 to <34 weeks)	7 (3.0%)
Late preterm (34 to <37 weeks)	21 (9.1%)
Term (≥37 weeks)	197 (85.7%)
Weight (grams)	3565 (3042, 3914)
Female gender n (%)	100 (44%)
Apgar	
1 min Apgar	5 (4, 7)
5 min Apgar	8 (6, 9)
10 min Apgar	9 (8, 10)
Umbilical cord values	
Arterial pH (*n* = 184)	7.20 (7.11, 7.25)
Arterial base excess (*n* = 176)	4.34 (1.67, 6.13)
Arterial pCO2 (*n* = 175)	8.16 (7.14, 9.64)
Pulse oximetry *n* (%)	222 (97%)
Positive pressure ventilation duration (seconds)	126 (65, 232)

## Data Availability

The data presented in this study are available on request from the corresponding author. The data are not publicly available due to privacy statements made in informed consent obtained from participants.

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
