# Peer review of "Use of Pulse Oximetry during Resuscitation of 230 Newborns—A Video Analysis"

_children, 2023, doi:10.3390/children10071124_

Round 1

Reviewer 1 Report

This study is a video analysis study on the use of PO in neonatal resuscitation and is judged to be a well-planned study. If the above-mentioned content is revised and reflected, it is considered to be publishable.

Page 1

Abstract

Line 17-18

“Newborn resuscitations were video recorded, and use 17 of PO during the first ten minutes of resuscitation recorded.” -> “Newborn resuscitations were video recorded, and use 17 of PO during the first ten minutes of resuscitation recorded and analyzed.”

Line 18-25

Results

Please add that video of the PO monitor signal was available in 118 (53%) of the 222 newborns.

Page 2

2.1. Setting

Line 72

Please check Reference 14.

2.2. Data collection and Equipment

It would be nice to show the contents of “Data collection and Equipment” in a schematic diagram (Figure) or video.

Page 3-4

Results

The subjects of this study were ≥28 weeks gestational age and the median gestational age and birth weight of Table 1 were 3565 g at 40 weeks. Were there few newborns between 28-37 weeks of gestation? Please check the contents above.

Page 6

Discussion

Please add a few more similarities and differences with the existing literature on the use of PO for neonatal resuscitation.

Conclusions

The purpose of this study is ambiguous. It would be great if you could simplify and clarify the purpose and revise the first sentence of the discussion and conclusions accordingly.

Thank you.

Minor editing of English language required.

Reviewer 2 Report

This is an interesting manuscript and it is obvious that the authors have ample experience conducting this type of study. I only have a few points to make:

-          On rows 44-45, ”ECG present HR more rapidly than PO” – I would rephrase that – it is entirely debatable that all ECG devices are capable of that, despite the results of various studies that only tested a limited number of devices

-          On rows 46-47, the correct name is International Liaison Committee on Resuscitation – please correct

-          In subsection 2.3, it is stated that only resuscitations recorded on video were included – for which cases was recording decided? Was it an active decision or there were instances when people simply forgot to start recording?

-          On row 110, the word ”foot” is mentioned – what exactly is the point of using postductal oxygenation in the delivery room?

-          On row 140, it is stated that 4-6 persons were involved at some point in placing the pulse oximetry sensor which I find preposterous. What were they doing?

-          I think a median duration of 60 seconds is manageable for neonatal resuscitation. The purpose of pulse oximetry is preventing hyperoxia on prolonged resuscitations, so I don’t necessarily regard the argument that “application was not completed by the time PPV ended” (row 148) as being against pulse oximetry, this aspect should be further explored in the Discussions section

-          What was the exact sequence of events during placement of pulse oximetry? Was the pulse oximeter turned on before the start of the resuscitation? Was the sensor already connected to the pulse oximeter or the connection was made after the sensor was placed on the infants’ limb? Was the palm of the hand or the wrist predominantly used? Was there a designated HCP to first place the pulse oximetry?

-          On row 176, the authors mention that ”PO signals may be delayed in newborns with low Apgar scores”. Did they explore a correlation between long periods to achieve PO signals and low Apgar scores, low oxygenation values or low gestational ages?

Reviewer 3 Report

Thank you for the opportunity to review this work. This manuscript demonstrates the usefulness of pulse oximetry used during newborn resuscitation. Detailed comments about this study are as follows:

-Figure 2 is an excellent illustration for a time of placement on the resuscitation table in the first consecutive ten minutes time frame to represent the PPV duration and application and/or adjustment of pulse oximetry (PO). According to the objective of this study, as the proportion of time with a useful PO signal during resuscitation, it should consider adding the new figure of the duration of a useful PO signal in the first consecutive ten minutes time frame.

-Owing to an observational design, it might have some potentially influential factors, such as gestational age, weight, and time spent applying the PO, that could affect the resuscitation and application of PO. It might consider performing the post hoc analysis to compare the time with a useful PO signal during resuscitation and some factors such as gestational age, weight, time spent applying the PO, etc.

-The author mentioned, "This study was limited by loss of PO signal data in 104 newborns due to technical challenges obtaining video signals." Please describe the technical challenge or common pitfall of obtaining video recording in this study. It might be helpful for other further research to manage this issue.

Round 2

Reviewer 3 Report

-The authors have modified the text following the reviewers' suggestions; therefore, I now have no further comments.